# A Review of Remote Sensing Approaches for Monitoring Blue Carbon Ecosystems: Mangroves, Seagrasses and Salt Marshes during 2010–2018

**DOI:** 10.3390/s19081933

**Published:** 2019-04-24

**Authors:** Tien Dat Pham, Junshi Xia, Nam Thang Ha, Dieu Tien Bui, Nga Nhu Le, Wataru Takeuchi

**Affiliations:** 1Geoinformatics Unit, the RIKEN Center for Advanced Intelligence Project (AIP), Mitsui Building, 15th floor, 1-4-1 Nihonbashi, Chuo-ku, Tokyo 103-0027, Japan; junshi.xia@riken.jp; 2Environmental Research Institute, School of Science, The University of Waikato, Hamilton 3240, New Zealand; hanamthang@huaf.edu.vn; 3Faculty of Fisheries, Hue University of Agriculture and Forestry, Hue 49000, Vietnam; 4Geographic Information System Group, Department of Business and IT, University of South-Eastern Norway, Gullbringvegen 36, N-3800 BøiTelemark, Norway; dieu.t.bui@usn.no; 5Department of Marine Mechanics and Environment, Institute of Mechanics, Vietnam Academy of Science and Technology (VAST), 264 Doi Can Street, Hanoi 100000, Vietnam; lenhunga70@gmail.com; 6Institute of Industrial Science, The University of Tokyo, 4-6-1 Komaba, Meguro-ku, Tokyo 153-8505, Japan; wataru@iis.u-tokyo.ac.jp

**Keywords:** coastal ecosystems, remote sensing, blue carbon, mangroves, seagrasses, salt marshes

## Abstract

Blue carbon (BC) ecosystems are an important coastal resource, as they provide a range of goods and services to the environment. They play a vital role in the global carbon cycle by reducing greenhouse gas emissions and mitigating the impacts of climate change. However, there has been a large reduction in the global BC ecosystems due to their conversion to agriculture and aquaculture, overexploitation, and removal for human settlements. Effectively monitoring BC ecosystems at large scales remains a challenge owing to practical difficulties in monitoring and the time-consuming field measurement approaches used. As a result, sensible policies and actions for the sustainability and conservation of BC ecosystems can be hard to implement. In this context, remote sensing provides a useful tool for mapping and monitoring BC ecosystems faster and at larger scales. Numerous studies have been carried out on various sensors based on optical imagery, synthetic aperture radar (SAR), light detection and ranging (LiDAR), aerial photographs (APs), and multispectral data. Remote sensing-based approaches have been proven effective for mapping and monitoring BC ecosystems by a large number of studies. However, to the best of our knowledge, this is the first comprehensive review on the applications of remote sensing techniques for mapping and monitoring BC ecosystems. The main goal of this review is to provide an overview and summary of the key studies undertaken from 2010 onwards on remote sensing applications for mapping and monitoring BC ecosystems. Our review showed that optical imagery, such as multispectral and hyper-spectral data, is the most common for mapping BC ecosystems, while the Landsat time-series are the most widely-used data for monitoring their changes on larger scales. We investigate the limitations of current studies and suggest several key aspects for future applications of remote sensing combined with state-of-the-art machine learning techniques for mapping coastal vegetation and monitoring their extents and changes.

## 1. Introduction

Blue carbon (BC) ecosystems consist of mangroves, seagrasses, and salt marshes, which play a crucial role across the world by providing habitats for wildlife and a range of ecosystem services to coastal organisms [1]. They are among the most carbon-dense ecosystems; while terrestrial forests are able to store 300 megagrams of carbon per hectare, seagrass meadows may store twice as much [2], and mangroves can store up to four times as much [3]. BC ecosystems also contribute significantly to the livelihood of coastal populations by providing valuable resources to local markets [4]. Despite such benefits and services, they have rapidly declined, owing to conversion to aquaculture and agriculture, overexploitation [5], and relative sea level rise [6].

According to the analyses of the available data provided by the Food and Agriculture Organization (FAO) in 2007, about 18.8 million hectares of mangroves existed worldwide in 1980 and declined to 15.2 million hectares by 2005. The most extensive mangrove area loss was recorded in Asia, followed by Africa, and North and Central America. Five countries, namely Indonesia, Australia, Brazil, Nigeria, and Mexico, account for 48% of the total global mangrove area. Just ten countries account for approximately 65% of the total global mangrove area, while the remaining 35% is accounted for by over 114 countries and regions, of which 60% have less than 10,000 ha of mangroves [7]. The comprehensive global assessment in [8] found that seagrasses have been consumed at a rate of 110 km${}^{2}$ per year since 1980, of which 29% have disappeared since seagrass areas were initially recorded in 1879. The regional-scale assessment [9] indicated that, in the U.S., the San Francisco Bay and New England have experienced a 79% and 50% reduction in the area of salt marshes, respectively.

Remote sensing-based approaches have been proven to be suitable for mapping and monitoring BC ecosystems [10,11]. They have lower costs, higher accuracy, and easier repeatability and cover wider areas than traditional field-based methods [11]. However, they still have limitations caused by clouds and limited coverage of airborne datasets [12]. The recent advances in remote sensing techniques, computer vision, and pattern recognition can overcome these limitations and encourage new approaches to develop more accurate mapping techniques [13]. In recent years, there has been an increased use of machine learning methods and data integration of optical and synthetic aperture radar (SAR) data in mapping and monitoring BC ecosystems.

In this paper, we inventory and give an overview and comparison of the key studies undertaken from 2010 onwards on mapping and/or monitoring of BC ecosystems. This review provides a critical overview of the methods developed from 2010 onwards using different remote sensing-based approaches and various datasets and specific mapping and monitoring techniques for mangrove, seagrass, and salt marsh ecosystems. The limitations of recent studies are also highlighted for future directions for the use of remote sensing techniques combined with state-of-the-art machine learning algorithms for mapping mangrove, seagrass, and salt marsh ecosystems and monitoring their extent. Importantly, this work also discusses the future trends in mapping and monitoring BC ecosystems.

## 2. Background and Methods

### 2.1. Blue Carbon Ecosystems

Coastal ecosystems mainly include mangroves, seagrasses, and salt marshes, which are known as “BC” [14]. They are capable of storing high volumes of carbon in their sediments [14]. They are able to contribute significantly to the global carbon cycle by absorbing and storing carbon for long time scales [2,15].

#### 2.1.1. Mangroves

Mangroves are forested wetlands and are mainly found in estuaries, along riverbanks, in lagoons, and in intertidal areas [16]. As shown in Figure 1, mangroves are groups of trees and shrubs that are able to grow in estuaries and form a transition zone between land and sea [15]. The mangrove biomes generally exceed half a meter in height and often grow above the mean sea level in the intertidal zone of coastal environments or along estuary margins of many tropical and semi-tropical countries around the world [17]. According to FAO, mangroves grow in the estuaries of over 120 countries [7] and cover over 137,000 km${}^{2}$ [18]. However, the most recent estimates indicate that between 1980 and 2005, the global areas of mangroves declined by up to 3.6 million hectares, mainly in Southeast Asia [7]. Primary drivers vary regionally, but include urban development, aquaculture, agriculture expansion, salt pond construction, and overexploitation [19].

#### 2.1.2. Seagrasses

Seagrass meadows are flowering, submerged monocotyledonous plants, e.g., tape grass and turtle grass, of tropical to temperate regions. They are usually found in shallow coastal water, have narrow grass-like leaves and often form dense underwater meadows [2]. Seagrasses are generally found in temperate and tropical climates (see Figure 2), and their meadows can survive for thousands of years. They provide several essential resources for aquatic life, including nursing and breeding grounds, water quality improvement, and coastline stabilization [21]. Recently, this ecosystem has been considered as an effective carbon sink, with a higher rate of carbon storage compared to Boreal and tropical forests [22]. Similar to mangroves, however, seagrass populations have declined [8], which has lead to a significant loss of aquatic habitats and the emission of CO${}_{2}$ to the atmosphere [23].

#### 2.1.3. Salt Marshes

Salt marshes are restricted to sub-tropical and temperate regions [1]. They are divided into intertidal marshes, which are found from the mean neap high water mark to the mean spring high water mark, and tidal marshes (see Figure 3), which are found above the spring high water mark [25]. Salt marshes are considered highly productive ecosystems because they provide a range of ecosystem benefits, such as food supply, nutrient cycling, and carbon storage [26]. Despite such important benefits, the area of salt marshes has been declining due to land development and dredging [27], as well as coastline transgression, i.e., sea level rise [28].

### 2.2. Inventory, Review, and Comparison of Studies

#### 2.2.1. Summary of Inventory, Datasets, and Methods

In this work, based on an extensive search in the Scopus and the Web of Science databases, more than 120 key studies conducted from 2010 onwards were inventoried. All inventoried papers were compared based on key attributes, including the data type and classification methods. All data types include the category, platform, spatial resolution, revisit capability, and launch date (Table 1). Figure 4 shows the locations of the key studies on a wide range of remote sensing applications for mapping and monitoring BC ecosystems.

Most of data used in the representative research is at high resolution and can be classified into four categories: (1) optical data, (2) SAR, (3) LiDAR, and (4) ancillary data. The ancillary data include maps produced from GIS data acquired by remote sensing and in situ measurements classified by their acquisition platform and sensor types. The dataset used in this review varies from different sensors, platforms, and spatial resolutions. We classified them into five categories: (1) optical high spatial resolution (HS), (2) optical medium/low spatial resolution (MS), (3) SAR, (4) LiDAR, and (5) ancillary data. The ancillary data include point, sonar, and video datasets and the spectral information collected by the field survey. Table 1 shows different datasets classified by their sensor types and acquisition platforms.

Among the available optical datasets, the Landsat series are the most widely used because they can be utilized for mapping for long-term observation at a large scale (e.g., global and country scales). LiDAR and HS datasets can be used to provide high precision mapping results. The SAR data can be used for both day and night acquisition. The spectral signatures derived from hyperspectral measurements could be used to distinguish the difference between mangrove, seagrass, and salt marsh species.

#### 2.2.2. Comparison of Studies

From Figure 4, it can be seen that the mangrove studies have been conducted all over the world, whereas the study sites of the seagrass are located mostly in South Asia and Europe. The study sites of salt marshes are located mainly in the USA, Canada, and Europe.

Figure 5 shows the classification methods for the mapping of BC ecosystems. In this work, we divided the classification methods into five types: (1) unsupervised learning; (2) supervised learning; (3) advanced learning; (4) object-based image analysis (OBIA); and (5) sub-pixel. Unsupervised and supervised learning methods belong to pixel-based classification techniques, whereby a pixel is assigned to a certain class by considering the similarities between the features. Typical unsupervised methods are the iterative self-organizing data analysis (ISODATA) technique and the index-based methods, e.g., normalized difference vegetation index (NDVI). Representative supervised methods are the minimum distance, the maximum livelihood classifier (MLC), and the decision tree (DT). Recently, machine learning methods, such as neural networks, support vector machines (SVMs), random forests (RF), and deep learning, have also been used in research fields. Here, we included these methods in the advanced category (3). The OBIA aims at analysing groups of neighbouring pixels as objects or segments instead of using the conventional pixel-based classification approaches. Image segmentation is a basic processing technique in the OBIA approach [30]. Most of these parameters are specified as objects and cannot be used in pixel-oriented classification [31]. In many cases, the results obtained with the object-oriented approach are similar to or outperform those obtained with the pixel-based approach. The sub-pixel approach exploits and identifies different materials within the single pixel using unmixing-based methods with the output of abundance. In the following three sections, we introduce mapping and monitoring change analysis for each BC ecosystem separately. In the mapping part, we depict the five categories of classification method and remote sensing datasets. In the monitoring part, we list the methods, remote sensing datasets, location, and the period of changes.

### 2.3. Mangrove Ecosystems

#### 2.3.1. Mapping Mangrove Ecosystems

There is a wide range of research studies available that have used different methods for mangrove mapping. This section reviews the advantages and disadvantages of each approach. Table 2 summaries the mangrove mapping techniques using remotely-sensed data from 2010–2018.

The most common and widely-used supervised classification method is the MLC [60]. Numerous studies have used the MLC for analysing land use and land cover (LULC) changes in both tropical forests and coastal zones [61,62,63]. This method is applied to mangrove mapping based on different types of remote sensing images, such as aerial photographs (APs) [64], Landsat TM [35,65], ETM+ [66], LANDSAT-8 OLI [67], IKONOS [38], SPOT [68,69], QuickBird [70], and RADARSAT [68]. The datasets obtained from Indian remote sensing satellite data [71] were also used to quantify mangrove loss for coastal zone management [33]. The MLC allows the extraction of different mangrove land cover categories [72]. The MLC technique exhibits an overall accuracy (OA) ranging from 79% to over 88% [62,63]. Furthermore, the MLC and the DT classifiers were used for mangrove mapping in South China [57,58] and Saudi Arabia with Landsat and Sentinel series [59].

For the unsupervised classification, the ISODATA classifier was applied to the IKONOS dataset acquired in Guinea [32], Landsat images from the Philippines [36] and Madagascar [51], and Indian Space Research Organization (ISRO)—Linear Imaging Self Scanner (LISS) satellite data from the Biosphere Reserve in Bangladesh [42] and South India [44]. The NDVI is an effective and the most popular index for vegetation monitoring [73], including mangrove ecosystem [74]. This index is often employed to obtain the spatial distribution of vegetation coverage in the wetland regions and mangrove ecosystems [74]. The use of the vegetation index leads to a better discrimination between mangroves and plantations [75]. Because vegetation indices are derived from remote sensing imagery, the spatial resolution of different imagery types affects the analytical results. Thus, the combination of a supervised classification method and texture analysis is a suitable approach to improve the accuracy assessment from various satellite imagery sources [76]. For instance, the MLC was utilized to discriminate mangrove and non-mangrove regions from extracted vegetation areas, which could be acquired based on the NDVI [76]. Giri et al. [41] and Jones et al. [55] used the supervised and unsupervised classification methods for the mangrove mapping globally and locally (within countries), respectively.

The object-oriented classification and the OBIA can provide 80% OA individually and can reach 94% accuracy for satellite image processing with Digital Elevation Model (DEM) thematic layers [30]. The object-oriented approaches for mapping mangroves have been applied in multi-resolution image segmentation. These techniques are also used for change detection conducted on the segmentation of different time series analysis. The object-oriented approaches are able to clearly discriminate between different land cover types within mangrove ecosystems. The methods can achieve an OA as high as 97% for the mangroves’ class and a kappa index of 0.83 [77].

In recent years, many researchers proposed a combination of techniques for multi-source remotely-sensed data. This method is called the hybrid approach, which combines both pixel-based and object-based classification techniques (see Figure 6) [78].

#### 2.3.2. Monitoring Mangrove Ecosystems

Time-series remotely-sensed data have been proved to be effective at monitoring mangrove ecosystems changes, using both APs and optical and SAR data. Table 3 lists the remotely-sensed data and the approaches used for monitoring mangrove ecosystems and assessing their extents and change detection. In the last decade, time-series APs have been used for mapping mangroves’ changes. Approximately eighteen studies conducted in nine countries in Asia, Australia, Africa, and North and South America have been reviewed. The most common techniques applied for classifying APs are based on visual interpretation and digitizing skills. Aerial photography has been used for detecting mangrove ecosystems changes and examining the drivers of these changes. For instance, Everitt et al. [64] used APs to detect mangrove dynamic changes due to an alternation of net accretion and erosion between 1951 and 1999, while Lam-Dao et al. [68] found a rapid expansion of shrimp aquaculture resulting in mangrove loss in the Ca Mau Peninsula (South Vietnam) from 1968–1992. A similar trend was observed using visual interpretation techniques for APs reported by [38] in Sri Lanka, showing that the development of shrimp farming is the main driving force of mangrove deforestation. The unsupervised ISODATA clustering algorithm was also used for processing aerial colour photographs taken to detect mangrove changes by Lam-Dao et al. [68], with high accuracy. The high spatial resolution derived from APs indicates their capability of mapping mangrove ecosystem extents and changes. Nevertheless, data acquisition and the presence of clouds, which is frequent in tropical and semi-tropical regions, are the main limitations of aerial photography for local mangroves’ mapping, change detection, and conservation and management support. Optical images such as the Landsat imagery are the most commonly used to evaluate mangrove change dynamics [66,69,79,80]. For instance, Nguyen et al. [69] and Bullock et al. [81] used the Landsat time-series data to monitor mangrove changes in the Mekong Delta, Vietnam. Data from the SPOT XS imagery have been used for monitoring mangrove dynamics in the tropics. Tran Thi et al. [82] used the SPOT time-series data to assess mangrove ecosystems’ changes in the Mekong Delta in Vietnam from 1995–2001. A recent study conducted by Liu et al. [83] demonstrated that SPOT and Gaofen-1 (GF-1) optical remotely-sensed data can be used for monitoring the temporal and spatial changes of mangrove species in the Mai Po area (Hong Kong). Jones et al. [55] employed the Landsat time-series data to monitor mangrove changes in Madagascar between 1990 and 2010.

Regarding the use of SAR data, Thomas et al. [84] compared the Japanese Earth Resources Satellite (JERS-1), SAR HH, and ALOS PALSAR HH and HV data for monitoring mangrove extents and concluded that a combination of two SAR datasets is suitable for assessing the mangrove extents over a decade and to support the JAXA’s Global Mangrove Watch (GMW) program. Pham et al. [85] reported that using the ALOS PALSAR data combined with the SVM algorithm have the potential to use the SAR data for monitoring forest extents and to support policy-makers in mangrove conservation and management.

### 2.4. Seagrass Ecosystems

#### 2.4.1. Mapping Seagrass Ecosystems

Table 4 summarizes the datasets and the mapping techniques used for mapping seagrass meadows. It is observed that a range of satellite sensors has been employed for mapping seagrasses, including optical data such as multi-/hyper-spectral imagery, SAR data, and APs. Among them, the multi-spectral imagery is the most frequently used. A series of research studies was carried out on HS multi-spectral imagery using WorldView and IKONOS for seagrass mapping. A comparison of the performances of different optical sensors, i.e., Landsat 8, Ziyuan-3A, Sentinel–2, and WorldView–3, on seagrass mapping was reported by Kovacs et al. [110]. The OBIA was used in this study to detect the boundaries of the seagrass species. Despite the very high spatial resolution of ZY–3A and WorldView–3, the OA only obtained 66% for species mapping and 57% for percentage cover maps. The highest user accuracy was reported using the WorldView-3 data, with 80% for seagrass species and 76% for seagrass percentage mapping (at 70–100% cover). A similar performance of WorldView-2 was observed for seagrass species and percentage cover mapping, in which the average producer accuracies were 73.74% and 75%, respectively [111]. An OA of 78% was obtained using WorldView–2 and OBIA for seagrass distribution mapping [112]. Wicaksono [113] used an image rotation technique, involving principle component analysis (PCA) and independent component analysis to deliver the spatial distribution of seagrass from WorldView–2 data. The water column corrected image with principle component analysis was reported to have the highest producer accuracy for seagrass mapping from the MLC. Using simple radiative transfer modelling for water column correction, the corrected WorldView-2 produced a seagrass map with 88.3% OA [114]. Similar research was conducted for mapping of seagrass distribution around Lizard Island, Australia [115] using the OBIA and the radiative transfer modelling-based simulator [116]. For the IKONOS sensor, the selected works were [117,118] with a remarkable density index developed by Baumstark et al. [119]. However, a low Kappa coefficient (0.53) was observed in this work due to the inconsistency of the index and validation data.

At a lower spatial resolution, a large-scale mapping is preferred to measure the effects of the natural hazards (in Indian [128] and Japan [144], for instance) or to provide more data for resource inventory with typical works in the lower Alaska Peninsula [136] and Landsat-5 TM in the Eastern Africa coast [165]. Other works involve seagrass mapping from Landsat-8 OLI [145] and [147]; ASTER, SPOT–4, and KOMPSAT–2 [138]; ALOS AVNIR–2 [120] and [152]; ALOS AVNIR–2 and ASTER using leaf area index [132]; Indian remote sensing (IRS) [125] and THEOSdata [135].

Although multi-spectral imagery has emerged as a popular dataset for seagrass mapping, the limited number of spectral bands may lead to a low accuracy of single species detection. For this reason, hyper-spectral imagery was widely combined with physical-based models and various classification algorithms to improve the accuracy of seagrass detection in complex water environment. A majority of studies used the applications of different hyperspectral sensors such as: CASI, HICO, HyMap, Hyperion, PRISM, and EO-1 images. In [148], the combination of CASI imagery and bathymetric LiDAR data for the detection of seagrass in shallow waters was reported. Four classification algorithms were tested, involving SVMs, MLC with principle component analysis (PCA-ML), MLC with linear discrimination analysis (LDA-ML), and spectral angle mapper (SAM). The data from the CASI imagery increased the OA to 95% for all habitats, whilst the fusion of CASI and LiDAR data only improved the accuracy of seagrass classification. Using the same hyper-spectral imagery for seagrass mapping, Valle et al. [139] applied MLC to six different band combinations of CASI imagery and yielded the highest producer accuracy using 10 bands’ combination (92%). A comparison of the performance was also found for Hyperion and a group of Landsat-5 TM, EO–1, and IKONOS [121,129], which exhibited a higher accuracy of seagrass mapping than hyper-spectral images. To deal with turbid waters, Borfecchia et al. [130] attempted to enhance the radiometric correction for 2.5-m spatial resolution airborne imagery. Landsat 7 ETM+ and MERIS images were used as the references for atmospheric and water column correction. In the study, the Lyzenga method for water column correction was improved with the addition of the water diffuse attenuation coefficient retrieved from MERIS images. A strong correlation ($R^{2}$ = 0.84) between the in situ leaf area index (LAI) and the water column corrected image pixel was achieved to deliver the seagrass map by the index. Despite promising results, there is still uncertainty on the method due to the spectral band mismatching of MERIS and the airborne images, which may lead to various results for water column correction.

In addition, Hedley et al. [153] introduced a physical inversion model combined with hyper-spectral data (PRISM) for the detection of seagrass species. Due to a similarity in the spectrum of seagrass (*Thalassia testudinum* and *Syringodium filiforme*) at the study sites, the proposed model failed to discriminate the species from only remote sensing data. This approach, however, is a potential option for other species with the addition of ecological data to the model. Another attempt to detect seagrass from algae using various data types and semi-analytical modelling approach was reported by Garcia et al. [140], who developed a semi-analytical shallow water forward model with hierarchical clustering technique from hyper-spectral (HICO and HyMap) and multi-spectral (WorldView–2) images. This work showed a better performance of HICO and HyMap compared to WorldView–2 in distinguishing seagrass from algae. These hyper-spectral data are able to detect seagrass with higher clustering accuracy and deeper water depth and can deal with various water types (Water Types 1–4). In addition to the spectral discrimination, this study also denoted the important role of spatial resolution and atmospheric correction for seagrass mapping and, in particular, distinction from algae. Similarly, Fearns et al. [122] reported an agreement of 48% for seagrass and up to 90% for brown algae by using semi-analytical shallow optical water modelling and HyMap data. It was considered that patchy meadows and water depth caused a low accuracy for the seagrass class when performing cross-validation with video data. In the same year, Lu and Cho [124] improved a water column correction algorithm for seagrass mapping. The improvement was then applied to APs and denoted an increase of reflectance values of the red and NIR bands of approximately 6% and 28%, respectively. The improvement of water column correction is expected to significantly contribute to further mapping of seagrass. Recently, the linear spectral unmixing classifier and visual interpretation [146] or an an extension of the LEGION method [127] were used to detect patchy seagrass meadows. Furthermore, Cho et al. [137] developed spectral modelling $Slope_{Red}$ and $Slope_{NIR}$ to detect seagrass from macroalgae by selecting the key bands from hyper-spectral imagery. Compared to SAM and ISODATA, the proposed models improved the accuracy to approximately 64%, which was higher than SAM (47.5%) and ISODATA (25%). Given their simplicity, $Slope_{Red}$ and $Slope_{NIR}$ may provide an alternative solution for the mapping of seagrass mixed with other substratum. Generally, the semi-analytical method using hyper-spectral imagery allows a higher mapping accuracy than the empirical approach [166]. However, it requires an intensive spectral library of different bottom curves as the input for the classification algorithm. This may lead to an expensive field sampling and storage of the library in case of large-area and mixed bottom type site monitoring. In addition, hyper-spectral sensors usually perform small coverage and allow on-demand requesting in a specific geographic region. The use of these data may result in the difficulty of time-series analysis and monitoring of seagrass resources at regional scales. In terms of physical-based models, it is necessary to validate the performance at various sites, as well as simplify the processing steps for further re-producibility .

Moreover, the modelling approach is constructed using digital images to predict the presence/absence of seagrass. The most recent work involved species distribution models using physical parameters and Bayesian geostatistics. Water depth, wave exposure, slope, and near-bottom velocity have been considered important factors in improving a model’s performance [134,143].

In the field of sonar images, the multi-beam bathymetry and backscatter data were combined to build a semi-automated sea floor mapping [126]. However, an accuracy assessment metric was not reported despite having a visual validation with the photo from the diving. On the contrary, Rahnemoonfar et al. [156] reported a very high accuracy (97%) for seagrass detection in turbid and very shallow water (<2 m depth) using morphology transformation and filter techniques on side scan sonar images. In [141], the authors compared the performance of single beam and QuickBird data for seagrass cover mapping. However, disagreement was observed in terms of both the Kappa coefficient and OA despite a good performance at high values of percentage cover.

Unmanned vehicles and machine learning are considered promising approaches in the marine science community. Unmanned marine vehicles (UMVs) have the advantages of both acoustic data from single beam echosounders and underwater cameras, which achieve very high accuracy in seagrass mapping (>95%) [155]. This high accuracy is also reported with unmanned aerial vehicles (UAVs) and the OBIA technique. Very high resolution photos were segmented with 61–100% producer accuracy for seagrass classification [158]. The potential combination of UAV/UMV and OBIA for seagrass mapping at a small scale was reported in previous studies. This approach represents a replacement for traditional seagrass field campaigns with snorkelling in the future [142]. In larger regions, however, the problems of high cost, time-series data, and the usage license of UAV/UMV may be an obstacle for seagrass mapping.

For machine learning-based mapping, several algorithms have been developed to improve either the classification from satellite imagery or seagrass prediction from environmental parameters. As described by Mohamed et al. [159], the weighted majority voting (WMV) method increased OA to 92.7% with the QuickBird image. A very high accuracy was also reported with the logistic model tree with digital images (96.33% incorrectness) [131], AdaBoost, Random forest (RF), Bayesian Network Learning, KNN, and NN (96% and 97% in both precision and recall) with digital images [150], or SVM algorithms (100% in producer accuracy) using the Sentinel-2 data [167]. A DT model was also adapted with hyper-spectral imagery for seagrass mapping [151]. The model performed well at high densities of seagrass meadows, but detected approximately 66% when the density reduced to below 60%. The authors claimed the threshold determination at deep optical water depths was the most challenging and led to the confusion of deep water and dense seagrass meadows. Regarding the prediction of seagrass distribution, RF was found to be superior to other machine learning models. In [149], a lowest RMSE (0.59) or highest precision and recall (98.1% and 90.4%, respectively) [160] were reported for the seagrass prediction model from environmental and physical factors (chlorophyll-a, distance to the coast). In the latter study, they also attempted to predict the presence of seagrass depending on their family. However, the performance was low in terms of both precision and recall (below 50% for all machine learning algorithms). Despite the importance of the idea, it is also necessary to address the uncertainty in the sampling method of the research. The disadvantage comes from an artefact of absent point data of seagrass, which can be different from their distribution in the field site. On the other hand, the unbalance of the dataset among seagrass families may cause a low accuracy of the prediction and lead to a re-evaluation of the accuracy assessment metrics in this case. A new approach is emerging, whereby the cloud computation power of Google Earth Engine, free satellite imagery, and machine learning have been combined to retrieve a map of seagrass and their change globally. Using Sentinel-2 and SVMs, approximately 2510 km${}^{2}$ of *Posidonia oceanica* in the Aegean sea, Greece, were mapped with an OA of 72% [161]. Despite the medium accuracy, it opens a modern approach for eco-regional mapping of seagrass unlike previous studies, in which a series of data (Landsat and ASTER) was combined with classical classification techniques for very large-scale mapping [133].

Deep learning algorithms provide a very comprehensive approach for marine science and, in particular, seagrass mapping. Among them, the deep neural network outperformed the others in the detection of fish [168], plankton [169], and coral reef [170] from digital images. Nevertheless, this approach is not widely used for satellite data, and it still faces big challenges in the application to seagrass mapping. As a result, a limited number of research papers was published on this subject in recent years. In this field, an automatic seagrass segmentation method was introduced that uses several feature extraction algorithms, involving convolutional neural networks (CNNs), histogram of oriented gradients (HOG), and local binary patterns (LBP) on digital images [162]. An excellent performance was found for the CNN method with an accuracy of 94.5% for seagrass segmentation. Despite the advantages of deep learning, the application for seagrass mapping remains a challenge due to mixed boundaries of seagrass and other benthic habitats, as well as the necessity of computation power. In addition, the effects of water columns on the image pixel values and the requirement of a very large dataset for the training phase will preclude the deep learning’s expansion to the marine science community [171].

#### 2.4.2. Monitoring Seagrass Ecosystems

Recently, several change detection approaches (see Table 5) have been applied to seagrass environments for various time scales, considering the change in spatial distribution, coverage, and above-ground biomass in the range of 5–40 years [172,173,174,175]. However, no single technique has emerged as superior across diverse marine environments [10].

Between 2010 and 2018, a series of post-classification-based change detection methods was developed. The cross-reference matrix was created from classified images to describe the change between 2009 and 2013 in Malaysia using Landsat 5 TM and Landsat 8 OLI [173]; between 2011 and 2016 in the Mediterranean Sea using RapidEye time-series data [174]; and between 2004 and 2007 using QuickBird and acoustic field data [172]. Linking satellite data to water quality, a time-series of MODIS images in five years (2007–2011) showed a negative correlation between the annual total seagrass area or biomass and water types with very high coefficients of 0.98 and 0.92, respectively [176]. For detection over a single year, a time-series analysis is a more suitable selection. However, it is sensor and acquisition date-dependent, and therefore, only a limited number of studies has been published in recent years [177]. For long-term monitoring of seagrass, the research papers selected involved change detection from 1996–2015 in Cam Ranh Bay (Vietnam), which used Landsat TM/ETM+/OLI [178], from 1992–2013 in Inhambane bay (Mozambique) [165], which used Landsat TM and SPOT-5, from 2004–2013 in the Eastern Banks, Australia [166], which used QuickBird–2, IKONOS, and WorldView-2, from 1990–2014 in Malaysia [179], which used Landsat TM/ETM+/OLI, and from 1972–2010 in Queensland, Australia [180], which used Landsat images. Similarly, the data of Landsat TM/ETM+, ASTER, SPOT-4, and Kompsat-2 were used to estimate the damage of seagrass meadows caused by a typhoon over 24 years in Korea. In this case, remote sensing data successfully detected much damage and provided a comprehensive understanding of seagrass dynamics at study sites [138]. However, as described in the research results of [181] and [179], the biggest disadvantage in long-term change detection is that an accuracy assessment was not conducted for all classified maps. This may lead to the uncertainty on seagrass area estimation and leave the problem of conservation zoning to the policy makers.

At a higher spatial resolution, APs have contributed considerably to mapping through the description of seagrass dynamics. Using visual interpretation and vector change analysis, the variation of the spatial distribution of seagrass was detected from APs in several regions and over several periods: in the northern Gulf of Mexico over 1940, 1971, and 2006 [182], in the Bay of Plenty, New Zealand, during the years 1959, 1996, and 2011 [183], and in Massachusetts, USA, over 1994–2007 [184]. Especially, very long-term change of seagrass from APs over 1939–2011 was also compared [181]. Despite the long period of change detection in this research, the accuracy of mapping was only reported for 2008–2011 due to the unavailability of ground truth data. By including modelling, Lyons et al. [177] developed a linear model for seagrass biomass estimation from benthic photos. This model was then reapplied to a time-series of remote sensing to retrieve the change detection of biomass in the past. Three models were built for biomass estimation, but they produced a low $R^{2}$ value. The highest $R^{2}$ achieved was 0.77, which was obtained when the above-ground biomass was a function of the *Syringodium isoetifolium* percentage cover. Object-based classification [135,166], clustering classification [125,138,178], and contextual editing with spatial analysis [135,182] are other interesting approaches for both high and very HS imagery in this category.

Machine learning has emerged as a new effective mapping approach to precisely detect the change of seagrass meadows over various time scales. At the time of this review, only one research paper has been published on Mediterranean seagrass [174]. In the study, using an RF algorithm, the inter-annual variation was evaluated from the RapidEye imagery with the OAs ranging between 73.5% and 82% over 2011, 2012, 2015, and 2016.

### 2.5. Salt Marsh Ecosystems

#### 2.5.1. Mapping Salt Marsh Ecosystems

The first work on salt marsh vegetation mapping using remote sensing data was reported by Budd and Milton [186], who used the first four Landsat TM data. Gao and Zhang [187] analysed the spectral characteristics of the salt marsh vegetation of four main regions in spring, summer, and autumn by a ground FieldSpec${}^{TM}$ and found that the discrimination ability in all regions was highest in autumn. Li et al. [188] applied multiple endmember spectral mixture analysis (MESMA) to AVIRIS images of salt marshes at China Camp in San Pablo Bay, California. Connel et al. [189] developed the tidal marsh inundation index (TMII) for daily MODIS 500-m surface reflectance data, which performed well in salt and brackish marshes in the Atlantic Ocean and Gulf Coast. The unsupervised (K-means) and supervised (MLC and SAM) classifiers were compared in [190] on different spatial resolutions of multi-spectral (IKONOS and QuickBird) and hyper-spectral images (ROSIS, CASI, and MIVIS). They found that: (1) hyperspectral images are superior to multi-spectral ones, and (2) the spatial resolution is more important than the spectral resolution. A similar investigation that led to similar conclusions was made in [191]. The machine learning-based techniques of vegetation community-based neural network classifier (VCNNC) [192], SVMs, and support vector data description (SVDD) [193] were used for the classification of salt marshes in tidal environments and in the European Union, respectively. These techniques achieved a better accuracy than MLC, and LiDAR and ground data played a crucial role in characterizing and classifying patterns of ground and low vegetation [194,195].

Table 6 summarizes the salt marsh mapping techniques using remotely-sensed data from 2010–2018. Because of the advanced remote sensing sensors and technologies used, most of the studies adopted multi-source and multi-temporal datasets or integrated different techniques. For a single dataset, Collin et al. [196] used the normalized difference LiDAR vegetation index model (NDLVIM) and SHOALStopography datasets to map intertidal habitats and their adjacent coastal areas (Gulf of St. Lawrence, Canada), and results with satisfactory accuracy were generated from a single LiDAR dataset using the NDLVIM and the digital terrain models (DTM) approach. Full-waveform LiDAR acquired at Cape Cod, Massachusetts, USA, was used for three marshes with five nonparametric regression methods, in which TreeNet’s stochastic gradient boosting produced the best results [197]. Connel et al. [189] developed the tidal marsh inundation index (TMII) for daily MODIS 500-m surface reflectance data, which performs well in salt and brackish marshes on the Atlantic Sea and Gulf Coast. Additionally, OBIA, SVMs, and RF classifiers have been widely applied on very high spatial resolution images (e.g., QuickBird and WorldView-2) [198,199,200] and hyperspectral images (e.g., AVIRIS and HyMap) [201,202] to examine the effectiveness of salt marsh mapping. The results demonstrated that the OBIA, SVMs, and RF classifiers were superior to the traditional classifiers [198]. High resolution SAR sensors, such as TerraSAR-X and Cosmo-SkyMed, are effective resources for mapping salt marshes in tidal flats and monitoring their seasonal variations. The best seasons for mapping and monitoring are winter and summer [203]. They also suggested the use of HH polarization in a single polarization for mapping salt marshes, because it can produce maximum backscattering.

For the multi-source dataset, Lucas et al. [207] integrated the datasets acquired from different optical sensors, namely the orthorectified SPOT-5 high resolution geometric (HRG) reflectance, ASTER, and IRS, as well as ancillary datasets to produce the first national map (Wales) of habitats in Europe through OBIA. For the combination of optical and SAR data, TerraSAR-X offers many ways to increase the performance of thematic mapping products, as well as L-band signatures provided by ALOS-PALSAR [208], along intertidal flats and coastal salt marshes using supervised classification methods [205]. Beijima [213] investigated the use of multiple sources (e.g., polarimetric SAR, elevation, and optical images) for the classification of salt marsh vegetation with RF. The RF classifier was found to be very powerful, and its performance was improved with the help of S-band and X-band SAR. For the integration of optical and LiDAR, Chust et al. [204] tested the discrimination potential of the LiDAR height and reflectance information, together with multi-spectral imagery (three visible and near-infrared bands), for the classification of 22 salt marsh and rocky shore habitats. The performance of LiDAR topographic variables and reflectance alone was poor (with OAs between 52.4% and 65.4%). The combination of the LiDAR-based DEM and derived topographical features with the near-infrared and visible bands achieved high OAs of between 84.5% and 92.1%. Airborne hyperspectral and LiDAR data were also used to map the salt marshes and riverbank vegetation based on multiple binary classification algorithms based on Fisher’s linear discriminant analysis (LDA) [206]. Similar studies were performed in [212,216]. In [211], spaceborne SAR and airborne LiDAR elevation (bare earth elevation and vegetation heights) were evaluated. The highest OA was achieved with SAR, LiDAR canopy, and the DEM data (81%), but no significant difference was observed from the SAR-only classification (81%). Both classifications exceeded the data combination using SAR data and DEM (66%) and SAR data with vegetation canopy (80%).

For the multi-temporal imagery, Reschke and Hüttich [214] developed an RF regression method based on multi-temporal Landsat data with HS datasets to extract sub-pixel information. Landsat-8 images (acquired in September and December 2013) were employed for mapping in coastal marshlands and mapping marshland using minimum distance, Mahalanobis, MLC, RF, and SVM classifiers. The highest performance was obtained using the MLC algorithms for the two Landsat-8 images (85.9%). Sun et al. [29] used time-series NDVI extracted from the Chinese HJ-1 optical images for the classification mapping and species identification of salt marshes.

#### 2.5.2. Monitoring Salt Marsh Ecosystems

Table 7 summarizes the change detection and monitoring techniques of salt marshes using various remote sensing datasets between 2010 and 2018. The areas studied and the periods of change are also described.

For monitoring salt marsh vegetation, publicly-available datasets, such as Landsat time-series, have been the most widely used. For instance, Sun et al. [227] proposed a flexible monthly NDVI time-series (MNTS) approach for multi-temporal salt marsh classification in the Virginia Coast Reserve, USA, by utilizing all viable Landsat TM/ETM+ images over the period 1984–2011 and indicated that the upper low marsh vegetation population significantly diminished in the analysed period. Jia et al. [224] investigated the changes of salt marshes in the Liao River Delta of China using Landsat TM time-series datasets acquired in 1988, 1995, 2000, 2004, 2007, and 2009. They used conversion matrices obtained from the classification tree to monitor the changes between salt marshes and other land cover types. Experiment results indicated that from 1988–2004, a larger area of salt marshes was replaced by other man-made land cover types and then was recovered from 2004–2009 by human activities. The authors of [219] mapped tidal flats and monitored the changes over very large areas using all Landsat Archive images and demonstrated their utility by mapping the tidal flats of China, the Democratic People’s Republic of Korea, and the Republic of Korea.

For several specific areas, historic maps, and HS data, including APs, airborne and sensor data, were used. Tuxen et al. [218] employed high-resolution (20 cm) remotely-sensed colour infrared imagery to map vegetation pattern changes of tidal salt marshes in the San Francisco Estuary over two years and performed a multi-scale analysis of derived vegetation pattern metrics. They also mapped six tidal marshes (two natural and four restored) in the San Francisco Estuary, CA, USA, between 2003 and 2004 using detailed vegetation field surveys and high spatial resolution colour infrared APs. They concluded that vegetation changes were significant, but the differences in composition and patterns across sites were larger than changes within sites over two growing seasons. Historical AP, HS satellite images, and GIS were used to quantify land cover changes in the inner section of Canal Principal, in the Bahia Blanca estuary [220]. Total losses of 33 and 6% of the area of mudflats and marshes, respectively, were observed, which may reflect increased erosion of relict Holocene coastal terraces in response to the rising sea level. Similarly, historical maps and APs taken from 1958–2010 were analysed to map salt marsh ecosystems and quantify LULC changes in the Alvor estuary and Arade River, Portugal [222]. They found that more than half of the salt marshes were lost due to dyke building and salt marsh reclamation for agriculture between approximately 1800 and 2010. In the mid-1960s, the abandonment of reclaimed agricultural areas resulted in the recolonization of salt marsh vegetation, which developed physically separated from natural marshes. Smith [223] explored the changes of six salt marshes within the Cape Cod National Seashore (CCNS) using a GIS-based methods with APs obtained from 1984 and 2013. Higher water levels could lead to significant changes. Multiple temporal high resolution images, such as QuickBird-2 and WorldView-2, were used to analyse salt marsh restoration in the Jamaica Bay, New York, based on OBIA. The study found that 21 hectares of salt marsh vegetation were lost between 2003 and 2013. Between 2012 and 2013, restoration efforts resulted in an increase of 10.6 hectares of salt marsh [227].

In addition, hyperspectral AVIRIS data taken over the Gulf of Mexico, USA, in September 2010 and August 2011 were used to assess the impact of oil spills on the salt marsh plant population with the change of vegetation index [221]. Furthermore, Beland et al. [225] developed field-referenced image endmembers and canonical discriminant analysis (CDA) to investigate the changes from 2010–2012. Marshes that were heavily contaminated with oil exhibited variable responses in this period. Marsh vegetation classes converted to subtidal and open water classes along oiled and non-oiled shorelines, respectively, that were similarly situated in the landscape.

A recent study reported by Da Lio et al. [228] showed that an advanced persistent scatterer interferometry (PSI) technique on a five year-long stack of X-band SAR acquisitions of the Venice Lagoon, Italy, can be used to quantify land subsidence in the salt marshes’ environment. They pointed out that land subsidence was much larger on man-made than natural salt marshes. However, to date, the number of studies using SAR data and a combination of SAR and optical data for monitoring salt marsh ecosystem is still limited. Thus, more studies on SAR and data fusion should be carried out in the near future for monitoring salt marshes.

## 3. Future Trends in Mapping and Monitoring BC Ecosystems

### 3.1. Future Trends in Mapping

This review indicated that optical imagery, such as multispectral and hyper-spectral data, is the most common for mapping BC ecosystems. Very HS imagery (e.g., WorldView, QuickBird, and IKONOS) is becoming more popular for small-scale mappings and can produce a better accuracy than high and MS imagery. For the classification of mangrove, seagrass, and salt marsh species ecosystems, hyperspectral imaging is effective and so is detection with the contribution of the spectral library. However, the hyperspectral data used were mainly airborne and constrained to limited areas. So far, SAR sensors have not been used commonly for mapping BC ecosystems despite the fact that SAR data can be acquired on larger areas in all weather conditions [11]. Thus, the integration of hyperspectral and SAR data offers important resources and further promotes the research studies in mapping BC ecosystems. A range of published research papers conducted atmospheric and water column corrections as a mandatory step in the image preprocessing for seagrass mapping. It can be concluded that very high spatial resolution imagery, such as SAR and LiDAR, and machine learning techniques are becoming more popular in dealing with patchy and mixed mangrove, seagrass, and salt marshes [229].

Despite the improved results in mapping BC ecosystems, a number of obstacles need to be further investigated, including: (i) the high cost of commercial sensors; (ii) the small area and low observing frequency of airborne, UAV, sUAV, and hyper-spectral imagery; (iii) the impact of the atmosphere and water depth on pixel values. Several potential solutions have been proposed, but there are still challenges with the detection of scatter meadows in turbid waters and with species detection. In terms of feature extraction, the OBIA has been effectively combined with very high spatial resolution imagery (<2 m). In many cases, the usage of feature extraction methods for medium spatial resolution (2–30 m) has not reached the expected accuracy (>85%). Therefore, OBIA training with machine learning algorithms should be addressed. On the other hand, it is necessary to motivate spectral library sharing to the public through the initiative of the open data gate. The spectral library data can strengthen the applications of hyper-spectral imagery, as well as improve existing models for seagrass species detection around the world.

The development of computer vision, pattern recognition, machine learning, and deep learning algorithms is expected to provide more effective tools in mapping BC ecosystems with promising results in the future. Several open source projects, which allow the integration of a wide range of machine learning techniques, such as the Python Scikit-learn library [230] and the Weka library [231], have been developed, significantly motivating the research community to improve the performance of machine learning algorithms for mapping BC ecosystems. The majority of qualified machine learning algorithms can be adapted to amend the detection under various conditions of marine and tidal environments. Finally, mapping should be standardized to unify the data sources for large-scale and comprehensive conservation around the world.

### 3.2. Future Trends in Monitoring

The change of ecosystem has been monitored using various satellite sensors over different time periods. The procedures are different from terrestrial change detection owing to the effects of water columns on pixel values. In almost all cases, satellite imagery must be pre-processed by various atmospheric and water column corrections, and therefore, the application of several change detection techniques is limited. The most frequently-used Landsat imagery has served as a very good multitemporal dataset for the monitoring of BC ecosystems. In addition, short-term and annual assessments have been improved owing to the availability of very high spatial resolution imagery. The Sentinel series, which has already been launched, provides an opportunity to monitor BC ecosystems using both optical (Sentinel-2A and -2B) and SAR (Sentinel-1A and -1B) images [232]. The impressive development of machine learning in the last few years has significantly contributed to the increase of the mapping accuracy, allowing more reliable monitoring. The combination of Landsat and a wide range of imagery, i.e., QuickBird, IKONOS, APs, UAVs, UMVs, and AUVs, provides a change detection assessment with higher spatial and temporal resolution. It is now important to consider the implementation of cloud computing such as the Google Earth Engine [233], which can automate and expand this task on a global scale.

However, in several cases, the combination of different sensors presents the challenges of the post-classification phase from the alternation in image processing and acquisition date of the sensors. The most common problem in change detection is the accuracy assessment of classified maps at different time points. Owing to the unavailability of data filed in the past, this task was not always performed in the studies considered in this review. As a result, the change evaluation is still questionable in many cases. An optimal technique for seagrass change detection has not been presented yet despite the development of several change detection techniques for land cover change [234]. Almost all research studies attempted to achieve the best accuracy of the mapping, and then applied a post-classification comparison (or image differencing) to retrieve the change assessment (classification-based approach). This approach is still reliable owing to the good performance of post-classification comparison [235], as well as the improvement of classification techniques.

In summary, future BC ecosystems monitoring research needs to consider the following points:Improving the mapping accuracy for each time point during the change detection period. Using multisource Earth observation data combined with state-of-the-art machine learning techniques, such as DT ensemble learning, i.e., RFs, rotation forests, and canonical correlation forests [236,237], may improve the mapping accuracy in certain periods.In case of a lack of ground truth data in the past, new accuracy metrics should be developed. Limited training data and data incompleteness are common in remote sensing, especially in large-scale time-series datasets. Further, fundamental technologies in remote sensing can deal with limited training data through novel detection techniques, such as transfer learning approaches with deep CNN for image classification [238,239].Addressing the best combination of multiple sensors with different techniques for change detection. Because each SAR and optical sensor has its own characteristics in reflecting BC ecosystems, the integration of different remotely-sensed data can offer a number of improvements in accuracy and data acquisition issues in monitoring BC ecosystems. However, the processing time over large areas involved in time-series image analysis should be taken into account [11]. Thus, more research on multiple sensor for monitoring BC ecosystems is needed in the future.Developing a standard framework for change detection assessment to enhance the reliability of change detection and to automate the image processing throughout the world. In this context, high performance computing (HPC) facilitates the process, and programming skills are required. This allows researchers to update automatically and re-use classification algorithms, making research faster and expanding the boundaries of BC research [240].Understanding the rate of change and the drivers of BC ecosystems. Quantifying the diversity of drivers of BC ecosystems’ changes is important for policy implementations for sustainable conservation and management all over the world [19]. Using combined multitemporal and multi-sensor data can help quantify the key drivers of coastal ecosystem changes.

## 4. Concluding Remarks

BC ecosystems are important to coastal communities, but continue to be threatened all over the world. As they cover relatively large areas, commonly inaccessible for field research, remote sensing is an alternative tool for mapping and monitoring their changes. This review highlighted significant contributions of remote sensing datasets and various techniques applied on BC ecosystems (e.g., mangrove, seagrass, and salt marsh). High spatial resolution data can improve the accuracy of mangrove, seagrass, and salt marsh classification. Medium spatial resolution data, such as the Landsat time-series, are the most widely-used data for monitoring BC ecosystems on larger scales. Active remotely-sensed data, such as SAR and LiDAR, can contribute to higher performance in mapping and monitoring ecosystems. Multi-temporal high spatial resolution images have been used to monitor the changes in specific areas. Incorporation of multi-resolution and multi-source (SAR, multispectral, and LiDAR) data may improve the monitoring accuracy.

A critical overview of the key studies undertaken from 2010 onwards on the most common mapping and monitoring techniques, as well as the remote sensing datasets, was presented in this work. Research efforts have been made with optical sensors, such as multispectral and hyperspectral datasets and different traditional methods for mapping and monitoring BC ecosystems. We gained several insights into the research trend for mapping and monitoring BC ecosystems. Slightly more attention seems to have been paid to the advanced methods or the hybrid methods using multi-source and multi-temporal datasets. In the near future, more advanced sensors, such as SAR and LiDAR, and novel machine learning approaches using ensemble DTs and deep learning methods should be used for the mapping and monitoring of BC ecosystems. Focus on the development and choices of state-of-the-art machine learning algorithms should be placed for mapping and monitoring in future studies.

## Figures and Tables

**Figure 1 sensors-19-01933-f001:**
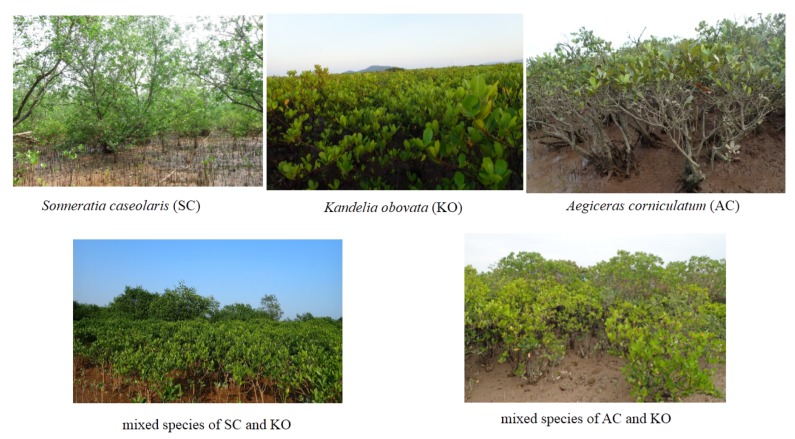
Mangrove communities in North Vietnam ($20^{\circ }$18${}^{\prime }$–$21^{\circ }$22${}^{\prime }$ N, $105^{\circ }$10${}^{\prime }$–$106^{\circ }$39${}^{\prime }$ E). These photos were taken by T.D. Pham [20].

**Figure 2 sensors-19-01933-f002:**
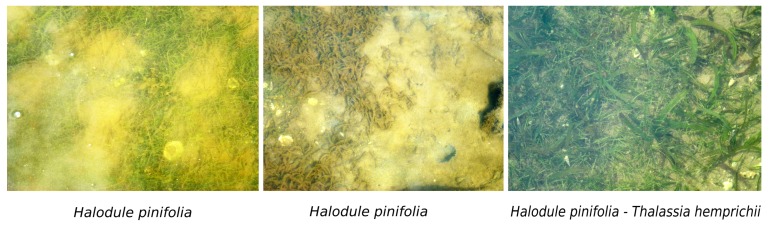
Seagrass meadows in the centre of Vietnam (16${}^{\circ }$13${}^{\prime }$–16${}^{\circ }$42${}^{\prime }$ N, 107${}^{\circ }$21${}^{\prime }$–108${}^{\circ }$5${}^{\prime }$ E). These photos were taken by H.N. Thang [24].

**Figure 3 sensors-19-01933-f003:**
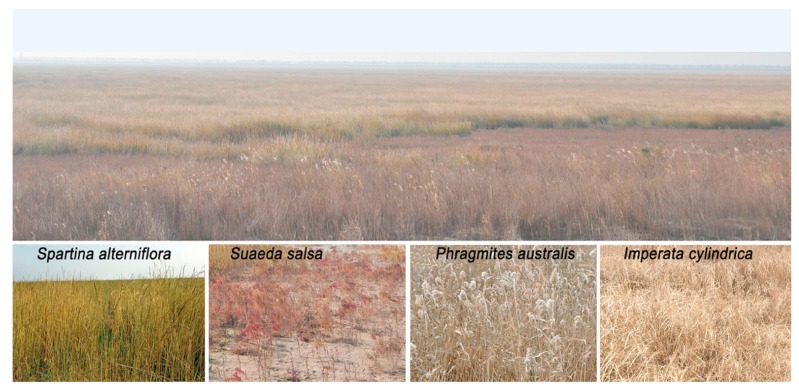
Salt marsh ecosystem in the middle coast of Jiangsu, China ($33^{\circ }$00${}^{\prime }$–$33^{\circ }$40${}^{\prime }$ N, $120^{\circ }$30${}^{\prime }$–$120^{\circ }$55${}^{\prime }$ E) [29].

**Figure 4 sensors-19-01933-f004:**
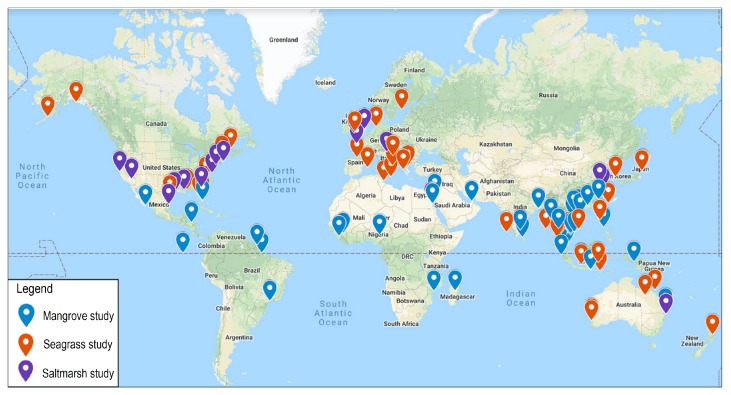
The review of the study sites.

**Figure 5 sensors-19-01933-f005:**
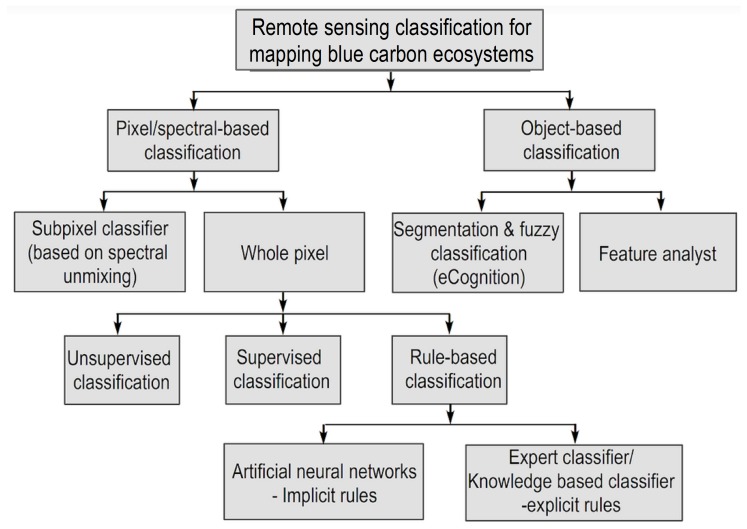
Illustration of classification methods for mapping blue carbon ecosystems.

**Figure 6 sensors-19-01933-f006:**
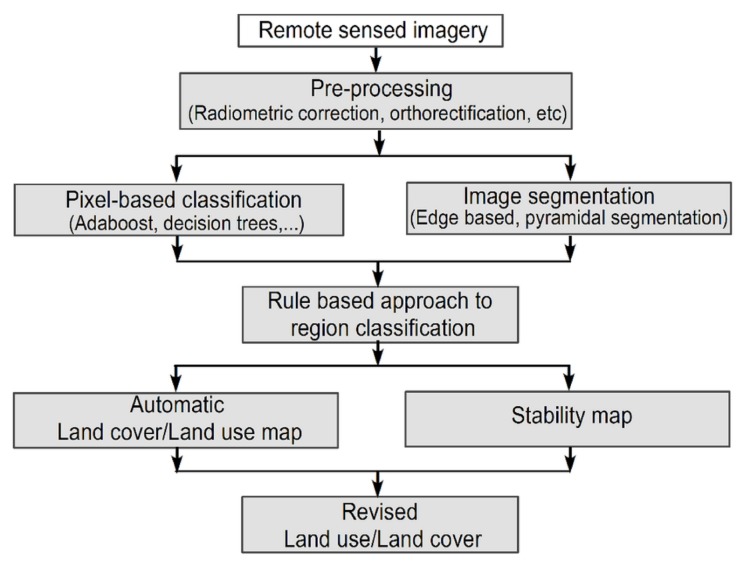
The hybrid approach scheme (modified from Zingaretti et al. [78]).

**Table 1 sensors-19-01933-t001:** Summary of remote sensing data used for mangroves, seagrasses, and salt marshes. HS, high spatial resolution; MS, medium/low spatial resolution.

Type	Acquisition	Platform	Spatial Resolution (m)	Revisit Capability (day)	Launch Date (year)
Optical HS	Airborne	UAV HyMap CASI	0.1 Hyperspectral: 5 Hyperspectral: 1	mobilized to order	Since 2000
	Spaceborne	QuickBird	Panchromatic: 0.6 Multispectral: 2.4	1.5–3	2001.10
		IKONOS	Panchromatic: 1 Multispectral: 4	1.5–3	1999.9
		ALOS	PRISM: 4 AVNIR: 10	2	2006.1
		SPOT-4	Panchromatic: 10	2–3	1998.3
		SPOT-5	Panchromatic: 5 Multispectral: 10	2–3	2002.5
		WorldView-2	Panchromatic: 0.46 Multispectral: 1.85	1.1	2009.10
		GeoEye-1	Panchromatic: 0.41 Multispectral: 1.65	2–3	2008.9
		KOMPSAT-2	Panchromatic: 1 Multispectral: 4	2–3	2007.7
Optical MS	Space borne	Landsat 5	Multispectral: 30	16	1984.4–2013.5
		Landsat 7	Panchromatic: 15 Multispectral: 30	16	1999.4
		Landsat 8	Panchromatic: 15 Multispectral: 30	16	2013.2
		Sentinel-2	Multispectral: 10, 20, 60	5–10	2015.6
		IRS1D LISS	Multispectral: 23.5	25	1997.9
		EO-1	Hyperspectral: 30 Multispectral: 30	16 200	2000.11
SAR	Space borne	ALOS	PALSAR: 10	46	2006.1
		ALOS-2	Spotlight: 1-3 Stripmap: 3, 6, 10	16	2014.5
		RADARSAT-2	Spotlight: 1 Stripmap: 3	24	2007.12
		Sentinel-1	Interferometric Wide Swath: 5 Stripmap: 5	12	2014.4
LiDAR	Ground Aerial	Aeroplane, UAV	0.1	mobilized to order	Since 2000

**Table 2 sensors-19-01933-t002:** Remote sensing techniques for mapping mangroves using remotely-sensed data from 2010–2018. OBIA, object-based image analysis.

	Datasets				Methods				Location
	HS	MS	SAR	LiDAR	SL	UL	OBIA	Advanced	
Kovacs et al. [32], 2010	*√*					*√*			West Africa
Salami et al. [33], 2010		*√*			*√*				Nigeria
Yu et al. [34], 2010		*√*						*√*	China
Alatorre et al. [35], 2011		*√*			*√*				Mexico
Long and Giri [36], 2011		*√*				*√*			Philippines
Satyanarayana et al. [37], 2011a	*√*					*√*			Sri Lanka
Satyanarayana et al. [38], 2011b	*√*				*√*				Malaysia
Beh Boon et al. [39], 2011		*√*						*√*	Malaysia
Dat and Yoshino [40], 2011	*√*	*√*					*√*		North Vietnam
Giri et al. [41], 2011		*√*			*√*	*√*			Global
Nandy and Kushwaha [42], 2011		*√*				*√*			Bangladesh
Heumann [43], 2011	*√*				*√*			*√*	Ecuador
Srinivasa Kumar et al. [44], 2011		*√*				*√*			India
Chadwick [45], 2011	*√*			*√*	*√*				Florida, United States
Rocha de Souza Pereira et al. [46], 2011			*√*				*√*		Brazil
Tien Dat and Yoshino [20], 2012	*√*		*√*				*√*		Hai Phong, Vietnam
Kirui et al. [47], 2013		*√*			*√*				Kenya
Vo et al. [48], 2013	*√*						*√*		Ca Mau, Vietnam
Cardoso et al. [49], 2014		*√*			*√*				Amazon, Brazil
Kamal et al. [50], 2014	*√*							*√*	Moreton, Australia
Jones et al. [51], 2014		*√*			*√*				Madagascar
Singh et al. [52], 2014		*√*						*√*	India
Kamal et al. [53], 2015	*√*	*√*				*√*			Karimunjawa, Indonesia
Giardino et al. [54], 2016		*√*			*√*				Myanmar
Jones et al. [55], 2016		*√*			*√*	*√*			Madagascar
Aslan et al. [56], 2017		*√*	*√*				*√*		Indonesian Papua
Chen et al. [57], 2017		*√*	*√*		*√*				China
Zhang et al. [58], 2017		*√*			*√*				China
Almahasheer [59], 2018		*√*			*√*				Arabian Gulf

**Table 3 sensors-19-01933-t003:** Remote sensing techniques for monitoring mangroves using remotely-sensed data from 2010–2018. MLC, maximum livelihood classifier.

Technique Used	Sensor	Location	Reference	Year of Detection	Year of Publishing
MLC	Aerial photographs	South Texas Gulf Coast, USA	[64]	1976, 1988, 2002	2010
MLC	Landsat TM	Madagascar	[65]	1951 and 2000	2010
MLC and ISODATA	Landsat, SPOT, and RADARSAT	Ca Mau Peninsular, Vietnam	[68]	1973–2008	2011
MLC	IKONOS	Sri Lanka	[38]	1994 and 2004	2011
Unsupervised	Landsat TM and IRS 1D LISS-IV	Chidambaram, South India	[44]	1991, 2001, 2006	2011
Visual interpretations	Landsat MSS, TM, and IRS LISS-III	East coast of India	[86]	1973, 1990, 2006	2011
Sub-pixel	MODIS	Mahakam Delta, Indonesia	[87]	2000–2010	2013
Unsupervised	Landsat TM, ETM+, and OLI	Honduras	[88]	1985-2013	2013
Unsupervised	Landsat TM and ETM+	Zhanjiang mangrove, Guangdong province of Southern China	[79]	1977–2010	2013
MLC	Landsat TM and SPOT	Kien Giang Province, Vietnam	[69]	1989–2009	2013
Unsupervised	Landsat, JERS-2 SAR, ALOS PALSAR, and ALOS-2 PALSAR-2	Global Mangrove Watch	[89]	1992–2011	2014
RoFand NN	Landsat MSS, TM, and ETM+	Ayeyarwady Delta, Myanmar	[90]	1978–2011	2014
OBIA	Landsat TM	Quang Ninh, Ca Mau, Kien Giang in Vietnam	[91]	1990, 2000, 2010	2014
SVM	Landsat TM	Southeast coast of India	[52]	1991, 2000, 2009	2014
OSTU	Aerial photos, Landsat MSS, TM, ETM+, and SPOT 2, 4, 5	Mui Ca Mau, Vietnam	[82]	1953–2011	2014
MLC	Landsat TM and ETM+	Southern Peninsular Malaysia	[66]	1989–2014	2015
OBIA	Landsat TM and ETM+	Matang Mangrove Forest Reserve, Malaysia	[67]	1988–2013	2015
CART	Landsat TM, ETM+, and OLI	South Asia	[92]	2000-2012	2015
MLC	Aerial photographs and Landsat	Mui Ca Mau, Vietnam	[93]	1953–2011	2015
OBIA and visual interpretation	ALOS PALSAR and JERS-1 SAR	Nine mangrove sites in Brazil, Australia, French Guiana, Kalimantan, Papua, Sumatra of Indonesia, Peninsular Malaysia, Nigeria, and Ecuador	[84]	1995–1998 and 2007–2010	2015
ISODATA	Aerial photos, ASTER, and Landsat ETM+	Ecuador	[94]	2000–2011	2015
OBIA	Landsat TM and OLI	Hai Phong city, Vietnam	[95]	1989–2013	2015
OBIA	Landsat TM, OLI	Ca Mau Peninsula, Vietnam	[96]	1979–2013	2015
Unsupervised	Landsat TM, ETM+, and OLI	Zambezi Delta, Mozambique	[97]	1994-2013	2015
Tasselled cap transformation (TCT) and subpixel	Landsat TM and OLI	Can Gio Biosphere Reserve, Vietnam	[98]	1989–2014	2016
NDVI	Landsat TM and ETM+	Mekong River Delta, Vietnam	[99]	1989–2014	2016
supervised and unsupervised	Landsat TM, ETM+, OLI	Southeast Asia	[19]	2000–2012	2016
MLC	IKONOS, GeoEye, QuickBird, and WorldView-2	Bali, Indonesia	[100]	2001–2014	2016
MLC	Aerial photos, QuickBird, and WorldView-2	Northeastern coast of Florida, USA	[70]	1942–2013	2016
Unsupervised	Landsat TM, ETM+, and OLI	Madagascar	[55]	1990–2010	2016
OBIA	SPOT 5	Ca Mau Peninsula, Vietnam	[101]	2004–2013	2017
OBIA and SVM	SPOT 4 and 5	Can Gio Biosphere Reserve, Vietnam	[102]	2000–2011	2017
ISOCLUST	Landsat ETM+ and OLI	Madagascar	[103]	2002–2014	2017
TCT and RF	Landsat TM, ETM+, and OLI	Mekong Delta, Vietnam	[81]	1990–2015	2017
MLC	IKONOS, QuickBird, and WorldView-2 and 3, GeoEye	Perancak estuary, Bali, Indonesia	[104]	2001–2015	2018
K-means	Landsat TM, ETM+, and OLI	Sierra Leone, West Africa	[105]	1990–2016	2018
Data fusion	ALOS PALSAR and Rapid Eye	Wadi Lehmy, Egypt	[106]	2007–2015	2018
MLC	Landsat MSS, TM, ETM+, and OLI	Coastline of Bangladesh	[107]	1976–2015	2018
Decision tree	Landsat TM and Landsat OLI	Fujian Province, China	[108]	1995–2014	2018
SVM	ALOS PALSAR and ALOS-2 PALSAR-2	Cat Ba Biosphere Reserve, Vietnam	[85]	2010–2015	2018
MLC	Landsat TM, ETM+, and OLI	Tanintharyi, Myanmar	[109]	1989–2014	2018

**Table 4 sensors-19-01933-t004:** Seagrass mapping techniques using remotely-sensed data from 2010–2018.

	Datasets					Methods				
	HS	MS	SAR	LiDAR	Ancillary	SL	UL	OBIA	Advanced	Subpixel
Sagawa et al. [120], 2010	*√*					*√*				
Meyer et al. [121], 2010	*√*	*√*				*√*				
Fearns et al. [122], 2011	*√*				*√*	*√*	*√*			
Knudby and Nordlund [117], 2011		*√*				*√*			
Ferreira et al. [123], 2012		*√*					*√*		
Lu and Cho [124], 2012	*√*						*√*			
Nobi and Thangaradjou [125], 2012		*√*				*√*				
Micallef et al. [126], 2012					*√*			*√*		
Li et al. [127], 2012		*√*						*√*	*√*	
Paulose et al. [128], 2013		*√*				*√*				
Pu and Bell [129], 2013		*√*					*√*			
Borfecchia et al. [130], 2013	*√*	*√*					*√*			
Massot-Campos et al. [131], 2013	*√*					*√*				
Wicaksono and Hafizt [132], 2013	*√*	*√*					*√*			
Baumstark et al. [119], 2013		*√*				*√*	*√*			
Tamondong et al. [114], 2013	*√*					*√*				
Torres-Pulliza et al. [133], 2013		*√*					*√*	*√*		
March et al. [134], 2013	*√*					*√*				
Nguyen et al. [135], 2013		*√*						*√*		
Hogrefe et al. [136], 2014			*√*			*√*	*√*			
Cho et al. [137], 2014		*√*								*√*
Saunders et al. [115], 2015	*√*							*√*		
Kim et al. [138], 2015	*√*	*√*				*√*				
Valle et al. [139], 2015	*√*					*√*				
Garcia et al. [140], 2015					*√*					*√*
Barrell et al. [141], 2015	*√*							*√*		
Roelfsema et al. [142], 2015	*√*							*√*	*√*	
Schubert et al. [143], 2015	*√*								*√*	
Sagawa and Komatsu [116], 2015		*√*							*√*	
Tsujimoto et al. [144], 2016	*√*					*√*				
Purnawan et al. [145],2016		*√*					*√*			
Koedsin et al. [111], 2016	*√*					*√*				
Uhrin and Townsend [146], 2016	*√*									*√*
Wicaksono [113], 2016		*√*				*√*	*√*			
Baumstark et al. [112], 2016	*√*						*√*	*√*		
Kakuta et al. [147], 2016		*√*				*√*	*√*			
Pan et al. [148], 2016	*√*			*√*		*√*	*√*		*√*	
Folmer et al. [149], 2016	*√*					*√*				
Bonin-Font et al. [150], 2016	*√*					*√*			*√*	
Pe’eri et al. [151], 2016	*√*					*√*				
Pu and Bell [118], 2017	*√*	*√*				*√*	*√*			
da Silva et al. [152], 2017		*√*				*√*			*√*	
Hedley et al. [153], 2017		*√*								*√*
Traganos et al. [154], 2017										
Ferretti et al. [155], 2017	*√*						*√*			
Kovacs et al. [110], 2018										
Rahnemoonfar et al. [156], 2018	*√*								*√*	
Topouzelis et al. [157], 2018			*√*					*√*		
Ventura et al. [158], 2018	*√*							*√*		
Mohamed et al. [159], 2018	*√*	*√*				*√*			*√*	
Effrosynidis et al. [160], 2018	*√*					*√*			*√*	
Traganos et al. [161], 2018			*√*			*√*				
Gereon et al. [162], 2018	*√*								*√*	
Duffy et al. [163], 2018	*√*						*√*	*√*		
Konar and Iken [164], 2018	*√*					*√*				

**Table 5 sensors-19-01933-t005:** Monitoring seagrass change detection using remote sensing techniques.

Technique Used	Sensor	Location	Reference	Year of Detection	Year of Publishing
GIS vector analysis, MLC, post-classification comparison	QuickBird	Moreton Bay, Australia	[172]	2004 and 2007	2011
Change vector analysis	Landsat	Bay of Plenty, New Zealand	[183]	1959, 1996, 2011	2011
Edge-detection, visual interpretation, GIS contextual editing	Airborne imagery	Gulf of Mexico	[182]	1940–2007	2011
Change vector analysis, GIS contextual editing	Airborne imagery, orthorectified digital image	New England, USA	[184]	1994–2007	2011
Segmentation, post-classification comparison	Landsat	Southeast Queensland, Australia	[180]	1972–2010	2012
MLC, post-classification comparison	IRS	Lakshadweep Islands, India	[125]	2000 and 2008	2012
OBIA, time-series analysis	Landsat TM, Landsat ETM+	Moreton Bay, Australia	[177]	1988–2010	2013
Water type and seagrass change modelling, post-classification comparison	MODIS	Queensland, Australia	[176]	2007 – 2011	2014
OBIA, Arithmetic, GIS Contextual Editing	Landsat TM, ALOS AVNIR-2, THEOS	Phu Quoc Island, Vietnam	[135]	2001–2011	2014
Post-classification comparison	Landsat	Spermonde Archipelago, Indonesia	[185]	1972–2013	2014
Unsupervised classification, principle component analysis	Aerial photo	Port Phillip Bay, Australia	[181]	1939–2011	2014
OBIA, post-classification comparison	QuickBird–2, IKONOS, WorldView–2	Moreton Bay, Australia	[166]	2004–2013	2014
Mahalanobis distance, pixel analysis, post-classification comparison	Landsat TM, Landsat ETM+, Aster, SPOT-4, Kompsat–2	Korea	[138]	1990–2014	2015
Seed pixel growing, post-classification comparison	Landsat-5 TM, Landsat-7, Landsat-8	Malaysia	[179]	1990, 2000, 2014	2015
Linear modelling	Landsat-5 TM, Landsat-8	Malaysia	[173]	2009 and 2013	2016
Clustering, pixel analysis, post-classification comparison	Landsat TM, Landsat ETM+, Landsat OLI	Cam Ranh Bay (Vietnam)	[178]	1996–2015	2016
Post-classification comparison	Landsat TM, SPOT-5	Inhambane bay(Mozambique)	[165]	199–2013	2017
Time-series analysis, random forest	RapidEye	Mediterranean	[174]	2011 and 2016	2018

**Table 6 sensors-19-01933-t006:** Salt marsh mapping techniques using remotely sensed data from 2010–2018.

	Datasets					Methods				
	HS	MS	SAR	LiDAR	Ancillary	SL	UL	OBIA	Advanced	Subpixel
Collin et al. [196], 2010				*√*		*√*				
Chust et al. [204], 2010				*√*		*√*				
Ouyang et al. [198], 2011	*√*					*√*		*√*		
Dehouck and Lafon [205], 2011	*√*		*√*				*√*			
Bertels et al. [206], 2011	*√*			*√*		*√*				
Lucas et al. [207], 2011	*√*	*√*			*√*				*√*	
Dehouck et al. [208], 2012	*√*		*√*			*√*				
Lee et al. [203], 2012			*√*				*√*			
Mishra et al. [209], 2012	*√*				*√*		*√*			
Timm and McGarigal [199], 2012	*√*								*√*	
Zhang and Xie [201], 2012	*√*							*√*	*√*	
Zhang and Xie [202], 2013	*√*							*√*	*√*	
Hladik et al. [210], 2013	*√*			*√*		*√*				
Allen et al. [211], 2013			*√*	*√*					*√*	
Hladik and Alber [212], 2014				*√*	*√*	*√*				
Kumar and Sinha [191], 2014	*√*	*√*				*√*	*√*			
van Beijma et al. [213], 2014	*√*			*√*					*√*	
Carle et al. [200], 2014	*√*					*√*			*√*	
Reschke and Hüttich [214], 2014		*√*							*√*	*√*
Rapinel et al. [215], 2015		*√*				*√*			*√*	
Sun et al. [29], 2016		*√*					*√*			
O’Connell et al. [189], 2017		*√*					*√*			
Rogers et al. [197], 2018				*√*			*√*		*√*	

**Table 7 sensors-19-01933-t007:** Change detection and monitoring techniques of salt marshes using remote sensing datasets.

Technique Used	Sensor	Location	Reference	Year of Detection	Year of Publishing
Vegetation change	High-resolution (20 cm)	San Francisco Bay, CA, USA	[217]	1990 and 2000	2011
Vegetation change and MLC	Aerial photos (20 cm)	San Pablo Bay, CA, USA	[218]	2003 and 2004	2011
NDVI change	Landsat TM and ETM+	East Asia	[219]	2000–2012	2012
Post-classification comparison	Aerial photos	Bahia Blanca estuary, Argentina	[220]	1967, 1996, and 2005	2013
Index change	Hyperspectral AVIRIS	Gulf of Mexico, USA	[221]	2010 and 2011	2013
Vegetation change	Aerial photographs and maps	Algarve, Portugal	[222]	1958–2010	2014
GIS-based mapping	Aerial photographs	New England, USA	[223]	1984 and 2003	2015
Post-classification comparison	Landsat TM	Liao River Delta, China	[224]	1988–2009	2015
Canonical discriminant analysis classification comparison	AVIRIS	Gulf of Mexico, USA	[225]	2010 and 2012	2016
OBIA change	QuickBird and WorldView	Jamaica Bay, NY, USA	[226]	2003–2013	2017
NDVI change	Landsat TM/ETM+	Virginia Coast Reserve, USA	[227]	1984–2011	2018
Persistent Scatterer Interferometry	X-band SAR	Venice, Italy	[228]	1984–2011	2018

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
