# Peer review of "A Review of Remote Sensing Approaches for Monitoring Blue Carbon Ecosystems: Mangroves, Seagrasses and Salt Marshes during 2010–2018"

_sensors, 2019, doi:10.3390/s19081933_

Round 1
Reviewer 1 Report
General observations:
1. Abstract: Authors are invited to precise quantitatively their results.
2. Introduction: very short and not describing the entire research procedures.
3. Study area and data: poor description of the area and data (check some details below).
4. Figures and Figures captions: not clear or not complete.
5. Concluding remarks section not determinant.
Title: Because of the paper authors have carried out their research during the last 8 years starting from 2010, therefore, they are highly invited to change the title by completing it by introducing as example: “A decade review of Remote Sensing Approaches for Monitoring Coastal Blue Carbon Ecosystems: Mangroves, Seagrasses and Salt Marshes” or “A review of Remote Sensing Approaches for Monitoring Coastal Blue Carbon Ecosystems: Mangroves, Seagrasses and Salt Marshes during 2010-2018”.
In §2.1. (Line 62): Mainly in this paragraph, authors must absolutely include more examples and literature about the “The difference among mangrove, seagrass and salt marshes”. Unfortunately, the paragraph is restricted exclusively to 3 photos. However, authors can include more photos for cases/locations shown in Figure 2 (for example).
Line 66-67. Sentence style to be improved in: Figure 1 shows the real photos of mangroves, saltmarshes and seagrasses. For example, authors can use: Figure 1 shows some real photos of mangroves, saltmarshes and seagrasses for the case (cases) of (location or locations) (put here the exact corresponding location or locations; region, country, latitude, longitude and altitude…). Figure 1 caption (idem Line 66-67).
In § 2.1.1 (Line 77): Text to be putten before Figure 2. Line (80-81); “Figure 2 shows the locations of the key studies undertaken…”. The exact locations are not clearly shown on the map! Therefore, authors must include a corresponding table exposing the exact locations’ labels (name, location, region, country, lat, long, altitude…).
Table 1 caption not complete: HS? MS? Acronyms must be defined before and not after (in this paper in § 2.2). Table 1 to be putten after text in the corresponding paragraphe (§2.2). Column Spatial resolution (resolution???). Multispectral: 10,20,60 ??? ca 0.1???
Line 99-100 (§ 2.2.): Phrase not clear “Figure 3 concludes the classification methods, which are suitable for mapping mangroves, seagrasses, and salt marshes.”
Figure 3 caption is not clear, in contradiction with citation phrase in line 99-100 “Figure 3 concludes the classification methods, which are suitable for mapping mangroves, seagrasses, and salt marshes.”. Maybe it is not completed.
Figure 4. For the upper frame “Remote sensing imagery ”, no need to photos or icons (not apporting additional or relevant information).
Table 6. Information not complet in (Row 4 ; column 1): “? ], 2011”.
Impression of incoherence or contradiction between phrases in (to be taken in consideration by the authors):
- Line 491-492-493 (§ 4.2.): “Before 2010, the following studies should be pointed out for monitoring salt marshes vegetation. For instance, Eastwood et al. [215] indicated that MSAVI and GEMI are the best indices to use for salt marsh vegetation cover monitoring…”
and
- Line 16-17-18 (Abstract): “The main goals of this review paper are to show an overview and to summarize the key studies undertaken after the year 2010 on a variety of remote sensing applications for monitoring coastal blue carbon ecosystems and the limitations of ...”
Paragraph (§5.2.) must absolutely be improved. Additional information and propositions must be added to enrich the review study and the future research studies.
Line 607: Future research should consider the following issues:
Line 608: “improve the mapping accuracy for each time points during the change detection period;”. At least some propositions and guidelines should be given.
Line 609: “develop new accuracy metrics in case of unavailability of the ground truth data in the past;” At least some propositions and guidelines should be given.
Line 610: “address the best combination of different techniques for change detection;” At least some propositions and guidelines should be given.
Line 611: “develop a standard framework for change assessment to enhance the confidence of changing map through the world;” At least some propositions and guidelines should be given.
Line 613: “understand the reason and drive factors for the change.” At least some propositions and guidelines should be given.
Author Response
Dear the Reviewer;
The authors would like to thank the reviewer very much for your valuable observations, useful comments and suggestions. Major changes to the manuscript text are highlighted. We have read these comments and revised the manuscript accordingly, point by point. Please, find the answers to your comments and suggestions in the attached PDF file.
Best regards
Tien Dat Pham and co-authors

Reviewer 2 Report
This paper needs extensive professional English editing.
Nearly every paragraph had nonstandard word choice and incorrect words were often used.
That being said, I think it is an excellent review of a very important topic in remote sensing and I believe the list of references could be useful to many researchers. If the many English errors are corrected I think this would be a fine paper.
Here are is a list of corrections: it is not a complete list.
Introduction
First 2 sentences, bad grammar
Last sentence structure is confusing
Line 66: Replace real, I suggest representative
Figure 2: A legend would be helpful so the reader does not have to remember the color code for mangrove, seagrass, and salt marshes. I recommend spending some more time on a proper map and not just using a Google Map app.
Line 85: Change European to Europe.
Table 1: Please define HS, MS and SAR in the table
List the units of the Spatial resolution
Year.Month is not standard notation for defining date.
Please make this clearer
Table 2
Remote the columns Ancillary and Subpixel since non of the studies contain them. A column describing general location such as country or region would be very helpful.
Line 168: Remove [Table. reftable22]
Table 3.
Make the width of each column match the data below it. [Reference], [Year], and [Technique] should be more narrow. [Location] and [Sensor] should be wider. Make Location and Sensor left justified.
Line 233: Sentence is confusing, please revise
Lines 256 to 259: Issues with grammar, clarity, and abbreviations exist in this section. Please correct.
Line 298: Issues with grammar and clarity. Please correct.
Line 351-355: Many Issues with grammar and clarity. Please correct.
Line 354: Replace ‘be’ with ‘is’
Line 361-364: Many Issues with grammar and clarity. Please correct.
Lines 408-415: Many Issues with grammar and clarity. Please correct.
Table 5
Left justify table.
Author Response
Dear Reviewer;
The authors would like to thank the reviewer very much for your valuable observations, useful comments and suggestions. Major changes to the manuscript text are highlighted. We have read these comments and revised the manuscript accordingly, point by point. Please, find the answers to your comments and suggestions in the attached PDF file.
Best regards
Tien Dat Pham and co-authors

Reviewer 3 Report
This is an important and timely contribution to the literature and of high value to practitioners. The methods are sound, however the entire manuscript needs to be re-read and revised for grammar and flow. I've provided examples below but am unable to point them all out as there are far too many. Before this manuscript can be considered for publication serious time and effort needs to be put in to revising for clarity. As it currently reads, the poor grammar is very distracting.
Title, abstract, introduction: redundant language referring to coastal blue carbon ecosystems. Remove the term coastal. By default, blue carbon ecosystems are coastal.
Abstract:
-1-2 lines on the multi-faceted value of blue carbon ecosystems, including other goods and services + biodiversity - while very important, it's not all about the carbon.
-saying "these ecosystems have been lost worldwide" implies they are all gone. Clarify language to indicate that in much of the world (but not all) these areas are being degraded and converted.
-"resulting in their carbon stocks loss" is not grammatically correct - awkwardly phrased
-lines 4-6: awkwardly written; which roles?; compared with all other forest ecosystems?; practical difficulties and cost-effectiveness is contradictory; monitoring coastal vegetation implies change detection (i.e., monitoring)
-lines 6-9: "the" coastal ecosystems? all of them?
-lines 9-10: provides an alternative tool to what?
-lines 10-12: why do you separate optical imagery from aerial photos and multispectral data? This is optical imagery.
-lines 13-14: dynamics = change, therefore "dynamic changes" is redundant
-line 14: what do you mean by "current"?
-line 15-16: clarify that this is a review dealing simultaneously with all blue carbon ecosystems
-lines 16-20: this is run-on sentence - split in to multiple sentences
Introduction:
Paragraph 1 - setting the stage -
-opening lines are sparse and vague
-are you talking purely about tropical, sub-tropical BC ecosystems? If so, specify
-you say they're found in "most" climates, where are they not found?
-considered the most important what on earth? ecosystems? says who? and why?
suggested opening paragraph:
-what blue carbon ecosystems are
-found throughout the world, including tropical, sub-tropical latitudes
-due to ES goods and services, critical to coastal populations
-important to world due to massive carbon stocks / climate change mitigation
-provide important habitat / biodiversity
-despite their value - in much of the world rapidly declining due to x, y and z (need reference for this!)
Paragraph 2
-the second paragraph is kind of all over the place - be consistent
-also, no need to provide much detail on BC ecosystems here
-if your focus is on tropical / subtropical blue carbon ecosystems (unclear if this is the case?) then why talk about San Francisco and New England? -noted that your study includes whole world
---suggested: get rid of this paragraph - end the previous paragraph with clarifying whether all BC ecosystems are under consideration or just tropical / sub-tropical
Paragraph 3 - the role of remote sensing
-poorly organized
-lower cost, speed and wider scale compared to what?
-why so much focus on mangroves in this paragraph? Should keep things broad - not address one BC ecosystem type and leave the others out completely
-suggested:
---RS is a proven technology for map and monitor BC ecosystems (cite studies)
---RS comes at lower cost, higher accuracy, easier repeatability, covers more ground, etc., than traditional field-based methods
---not without limitations, e.g., clouds, limited coverage of airborne datasets, etc.
---technological advances continue - e.g., new data types, data combinations, computing capabilities, classification algorithms, etc.
Paragraph 4 - the aims of this paper
-suggested:
-this paper inventories, overviews and compares all studies using RS to map and / or monitor BC ecosystems from 2010 - onwards
-for overviews of efforts prior to 2010 see x, y and z
-limitations of recent studies highlighted - setting stage for future work
-you shouldn't have to tell the reader the standard order of sections! e.g., that your methods flow in to your results - in to your discussion - in to your conclusion. This is a given.
Brief overview: This sub-header should me changed
2 Background and Methods
2.1 Blue Carbon Ecosystems
Then elaborate on what these are and how they are different.
2.1.1 mangroves
---mangroves are defined as x
---according to x the world had x mangroves in x time in x countries
---by x time, this distribution had been reduced by x
---loss has occurred primarily in x, y and z
---primary drivers include x, y and z
-then take the same approach for seagrasses (2.1.2) and saltmarshes (2.1.3), telling the reader what each blue carbon ecosystem type is, where / how much + how much change, where most loss is occurring + why
Figure 1 - what is the origin of these photos? Where are these photos? Why do these single snapshots speak for all blue carbon ecosystems? Craft a figure which provides more examples of each - at least a 9 panel figure providing 3 examples of each blue carbon ecosystem type and capturing variability in appearance.
2.2. Inventory, Review and Comparison of Studies
In this paragraph you want to tell us your methodological approach
-clarify that you inventoried (first), reviewed, and then compared 120 studies
Figure 2: Needs a key - the caption should not tell the reader what the different colors mean.
What was 2.2 Summary of datasets and methods should become a part of section 2.2:
The meat of this section is fine - it requires a revision for grammar / flow to make it even clearer.
Suggested flow:
-based on internet-search/contacting authors, 120 studies from 2010 onwards were inventoried
-all inventoried papers were compared based on key attributes, including data type and classification method
-data type includes category, platform, spatial resolution, etc, (Table 1)
---then get in to some of the notable differences between these + refs
-classification methods include - list categories (Figure 3)
---then get in to some of the notable differences between these + refs
Then for each BC ecosystem type you go in to your separate sections, but I would re-order as below. Monitoring implies multiple observations over time / assessing dynamics and is therefore distinct from Mapping. Please be consistent. For each section / BC ecosystem type, follow a consistent formula outlined in previous section regarding the specific criteria being used for description and comparison -
2.3 Mangrove Ecosystems
2.3.1 Mapping Mangrove Ecosystems
2.3.2 Monitoring Mangrove Ecosystems
2.4 Seagrass Ecosystems
2.4.1 Mapping Seagrass Ecosystems
2.4.2 Monitoring Seagrass Ecosystems
2.4 Saltmarsh Ecosystems
2.4.1 Mapping Saltmarsh Ecosystems
2.4.2 Monitoring Saltmarsh Ecosystems
The Tables in these sections are great (!) - they really help demonstrate the inventory, description and comparison (i.e,. Tables 2, 3, 4, 5, 6, 7)
Table 2: Monitoring or Mapping? Be consistent
Table 3: lead with study as first column, then year in order (organized alphabetically by study within each year), then follow with criteria. Likewise for Tables 5, 7.
Future trends in mapping + monitoring
-don't use bullets in this section, use numbered points in a sentence
Author Response

(The authors gave the same response as above.)

Round 2
Reviewer 1 Report
- Authors' great efforts and real intention to improve the quality of their research paper and review is clear and appreciated. The revised manuscript is visibly showing the authors considerable work in carrying out the process of answering almost all the referees’ questions and concerns.
The revision and adjustment clarifies almost all the points I personally raised and helps me (and hopefully readers) understand the current manuscript vision, relevance and prominence.
In conclusion, although we appreciate the authors' efforts, I disagree with some of the conclusions drawn and mainly the omission of simple but objective and constructive remarks and additions. The following please find the points I think the authors may still take SERIOUSLY into account:
- Reconsider all tables and figures captions and style (to be homogenized). For example, in Figure 1. you have 5 pictures to be labeled (a, b, c, d and e). In the corresponding caption, in addition to the titile: Mangrove communities in North Vietnam: (a)……., (b)…. and so on for all similar figures and tables. In my previous review, I asked for the addition of: (the exact corresponding location or locations; region, country, latitude, longitude and altitude…).
- Figure 2. IDEM FIGURE 1. In addition, in my previous review I asked for: “…. authors must include a corresponding table exposing the exact locations’ labels (name, location, region, country, lat, long, altitude…).
- Figure 3. IDEM FIGURE 1 and FIGURE 2.
- Figure 4. I am suggesting an additional table with exact locations.
- In the last Concluding remarks paragraph: “…A comprehensive review of the most common mapping and monitoring techniques, as well as the remote sensing datasets, is presented in this work. Research efforts have been made with different types of datasets and methods…”. This sentence gives us the impression that the review has been carried out in general and for a very long period. However, the review study concerns only a limited period of 8 years (2010-2018). Therefore, authors are highly invited to take in consideration this observation.
- During the first round, as referee I tried to be subjective in considering just the significance of the scientific material and not the grammar, English, formatting…In total accordance with my colleagues the referees of this paper, even I don't feel qualified to judge about the English language and style, authors must absolutely do an effort to improve English language and style of their paper.
Author Response
Dear reviewer;
We appreciate very much the reviewer for your valuable comments and suggestions on our manuscript. Please find our response to each comment in the attached file.
Sincerely yours
Tien Dat Pham and co-authors
